# Human parvovirus B19 interacts with globoside under acidic conditions as an essential step in endocytic trafficking

**Jan Bieri**, **Remo Leisi**, **Cornelia Bircher**, **Carlos Ros** *

Department of Chemistry, Biochemistry and Pharmaceutical Sciences, University of Bern, Bern, Switzerland

* carlos.ros@dcb.unibe.ch

**Data Availability Statement:** All relevant data are within the manuscript and its Supporting information files.

## Abstract

The glycosphingolipid (GSL) globoside (Gb4) is essential for parvovirus B19 (B19V) infection. Historically considered the cellular receptor of B19V, the role of Gb4 and its interaction with B19V are controversial. In this study, we applied artificial viral particles, genetically modified cells, and specific competitors to address the interplay between the virus and the GSL. Our findings demonstrate that Gb4 is not involved in the binding or internalization process of the virus into permissive erythroid cells, a function that corresponds to the VP1u cognate receptor. However, Gb4 is essential at a post-internalization step before the delivery of the single-stranded viral DNA into the nucleus. In susceptible erythroid Gb4 knockout cells, incoming viruses were arrested in the endosomal compartment, showing no cytoplasmic spreading of capsids as observed in Gb4-expressing cells. Hemagglutination and binding assays revealed that pH acts as a switch to modulate the affinity between the virus and the GSL. Capsids interact with Gb4 exclusively under acidic conditions and dissociate at neutral pH. Inducing a specific Gb4-mediated attachment to permissive erythroid cells by acidification of the extracellular environment led to a non-infectious uptake of the virus, indicating that low pH-mediated binding to the GSL initiates active membrane processes resulting in vesicle formation. In summary, this study provides mechanistic insight into the interaction of B19V with Gb4. The strict pH-dependent binding to the ubiquitously expressed GSL prevents the redirection of the virus to nonpermissive tissues while promoting the interaction in acidic intracellular compartments as an essential step in infectious endocytic trafficking.

## Author summary

The neutral glycosphingolipid globoside (Gb4) has been historically considered the cellular receptor of B19V, however, its wide expression profile does not correlate well with the restricted tropism of the virus. Here, we show that Gb4 is essential for the infection at a step following virus uptake and before the delivery of the viral ssDNA into the nucleus. B19V interacts with Gb4 exclusively under acidic conditions, prohibiting the interaction on the plasma membrane and promoting it inside the acidic endosomal compartments, which are engaged by the virus and the GSL after internalization. In the absence of Gb4,

**Funding:** This study was supported by a grant from the Swiss National Science Foundation (grant 31003A_179384 to J.B.). www.snf.ch. The funders had no role in study design, data collection and analysis, decision to publish, or preparation of the manuscript.

**Competing interests:** The authors have declared that no competing interests exist.

incoming viruses are retained in the endocytic compartment and the infection is aborted. This study reveals the mechanism of the interaction between the virus and the glycosphingolipid and redefines the role of Gb4 as an essential intracellular partner required for infectious entry.

## Introduction

Parvovirus B19 (B19V) is a human pathogen discovered in 1974 [1]. The virus causes infections worldwide that vary in severity depending on the age as well as the immunologic and hematologic status of the host [2,3]. In healthy children, B19V causes a mild disease named *erythema infectiosum* or fifth disease [4]. The virus can occasionally lead to more severe complications, such as arthropathies in adults [5], *hydrops fetalis* in pregnant women [6] and chronic red cell aplasia in patients with underlying hemolytic anemia [7,8]. The linear single-stranded DNA genome of 5.6-kb in length is encapsidated within a small, non-enveloped, icosahedral particle consisting of 60 structural proteins, VP1 and VP2 [9]. These proteins share the same sequence except for an additional amino-terminal VP1 unique region (VP1u) of 227 amino acids. VP1u harbors strong neutralizing epitopes and is crucial to elicit an efficient immune response against the virus [10]. The two most relevant domains in the VP1u is a receptor binding domain (RBD) required for virus uptake into host cells [11] and a phospholipase A2 (PLA$_2$) domain required for the infection [12–15], presumably to promote endosomal escape [16].

B19V is transmitted primarily via the respiratory route [5]. From the respiratory epithelium, the virus particles access the bloodstream by an unknown mechanism. A striking feature of B19V is its marked tropism for erythroid progenitor cells (EPCs) in the bone marrow [17–19]. The lytic replication of the virus in this cell population accounts for the hematological disorders typically associated with the infection. The distribution of specific cellular receptors in concert with essential intracellular factors explains the remarkable narrow tropism of B19V [19–22].

Historically, the neutral glycosphingolipid (GSL) globoside (Gb4) or P antigen has been considered the primary cellular receptor of B19V [23]. Gb4 is expressed in target EPCs and the virus exhibits hemagglutinating activity, which can be inhibited by soluble or lipid-associated Gb4 [24,25]. The rare persons lacking Gb4 (p phenotype) are naturally resistant to the infection and their erythrocytes cannot be hemagglutinated by the virus [26]. Despite the solid evidence, the role of Gb4 as the cellular receptor of B19V has been increasingly questioned. The restricted tropism of B19V [17–19] does not align well with the wide expression profile of Gb4 [27,28]. Gb4 is the most abundant neutral GSL expressed on the membranes of human red blood cells (RBCs) [29,30], which cannot be productively infected. Moreover, the degree of virus attachment to cells does not correspond with the expression levels of Gb4, and although the presence of Gb4 was shown to be essential for the infection, it was not enough for productive infection [31].

The fact that the virus cannot internalize certain cells despite the expression of Gb4, suggests that other receptor molecules must be required for the uptake of the virus into susceptible cells. In line with this assumption, we showed that VP1u is required for virus uptake and identified a functional RBD at the most amino-terminal part of the protein, which mediates virus uptake independently of the rest of the capsid [11,19,32]. Different from Gb4, the expression profile of the VP1u cognate receptor (VP1uR) corresponds to the narrow tropism of B19V, limiting virus internalization and infection exclusively in cells at erythropoietin-dependent

erythroid differentiation stages [19,33]. Although VP1u is not accessible in native capsids, interaction with surface receptors on susceptible cells renders VP1u accessible [34,35]. This process could be partially reproduced by incubation of native capsids with soluble Gb4 [36], suggesting that the neutral GSL may assist the uptake process. However, in a recent study, we showed that in susceptible erythroid cells expressing VP1uR and lacking Gb4, VP1u becomes exposed and the virus is internalized, highlighting the irrelevance of Gb4 in this process. However, Gb4 was found to be essential for the internalized virus to initiate the infection [37].

It remains unclear whether B19V requires Gb4 as a host cell binding partner, or indirectly as a signaling molecule or cellular component required for the infection. The hemagglutination of human erythrocytes by B19V [24], which express large quantities of Gb4 [29,30], and the hemagglutination inhibition in the presence of soluble Gb4 [25], strongly indicate that B19V interacts with Gb4. However, attempts to confirm the interaction have not yet been conclusive. The binding of B19 virus-like particles (VLPs) to Gb4 in supported lipid bilayers has been reported [38], and the complex has been observed by cryoEM image reconstruction [39]. However, other studies using a higher resolution cryoEM failed to confirm the interaction [25]. In the same study, no binding signals above background controls were detected in various highly sensitive assays employing fluorescence-labeled liposomes, radiolabeled B19 capsids, surface plasmon resonance, and isothermal titration microcalorimetry [25]. These inconsistent observations suggest that either the interaction between B19V and Gb4 does not occur or requires specific conditions that have not yet been identified.

In this study, we have addressed the interplay between B19V and Gb4, the conditions required for the interaction, and the infection step where the GSL is essential. To this end, the hemagglutinating activity and adsorption capacity of native virions and virus-like particles (VLPs) to Gb4 expressed on human erythrocytes and erythroleukemia cells were examined under different experimental conditions. The role of Gb4 was investigated by following the infection of native virus and engineered capsids, differing in their capacity to interact with Gb4, in wild-type and Gb4 knockout erythroid cells. The study confirms the binding of B19V to Gb4, identifies the strict conditions modulating the interaction and redefines the essential role of the GSL at a post-internalization step for the infectious trafficking of the incoming virus.

## Results

### Gb4 is essential at a post-internalization step and before the delivery of the viral genome into the nucleus for replication

In an earlier study, we established a Gb4 knockout (KO) UT7/Epo cell line and demonstrated that Gb4 is essential for productive infection, but it is not required for virus attachment and internalization, which is mediated by the VP1u cognate receptors (VP1uR) [32,37]. Here, we used a recombinant VP1u construct (S1A and S1B Fig) to demonstrate that VP1uR expression is not altered in Gb4 KO cells. The expression of VP1uR in cells was sufficient to trigger virus attachment and uptake, however, in the absence of Gb4, the intracellular capsids failed to initiate the infection (Fig 1A and S2 Fig). In sharp contrast, blocking VP1uR by pre-incubation of cells with recombinant VP1u abolished attachment and internalization (Fig 1B).

To test the capacity of VP1uR to mediate virus internalization without the involvement of Gb4 or any other additional receptor(s) or attachment factor(s), recombinant VP1u subunits were chemically coupled to bacteriophage MS2 capsids, as previously described with some modifications [19] (S1C and S1D Fig). The engineered capsids were incubated with wild-type (WT) or with Gb4 KO UT7/Epo cells for 1h at 4˚C for virus attachment or at 37˚C for virus internalization. Regardless of the presence or absence of Gb4, the VP1u decorated particles

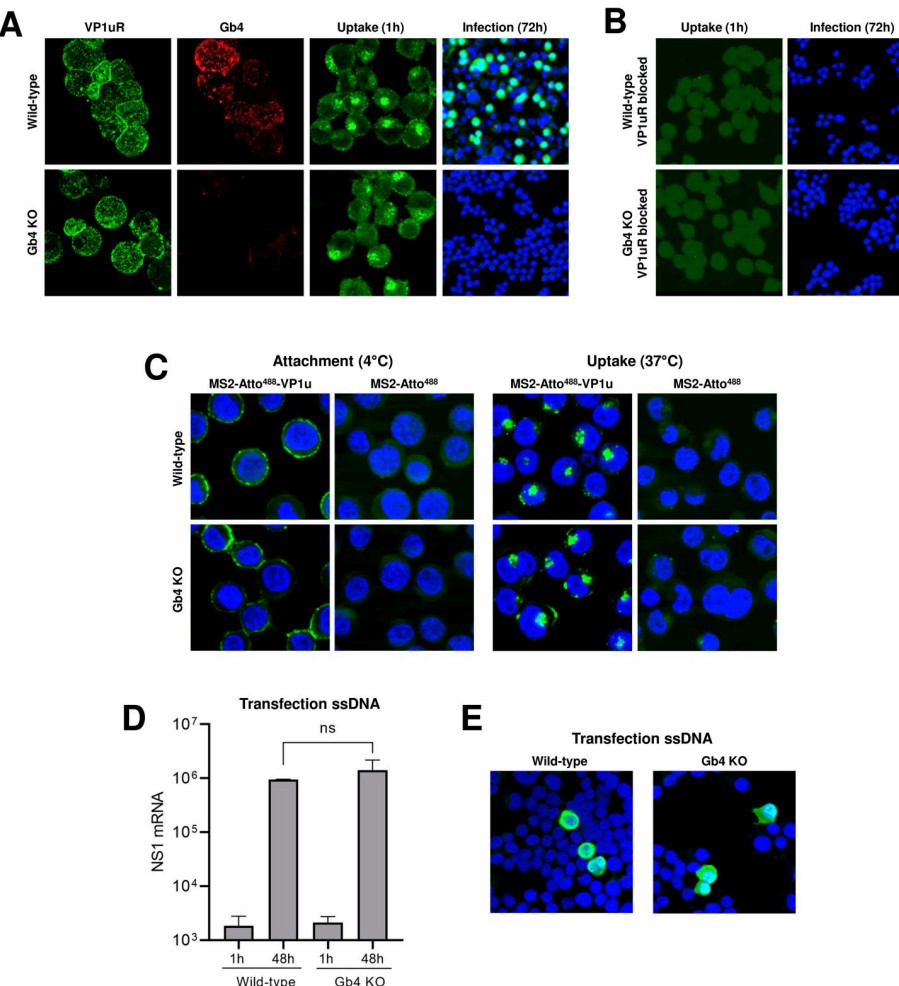

**Fig 1. Gb4 is essential at a post-internalization step and before the delivery of the viral genome into the nucleus for replication.** (A) Detection of VP1u cognate receptor (VP1uR) and Gb4 in UT7/Epo wild-type (WT) and Gb4 knockout (KO) cells by immunofluorescence (IF). Cells were incubated with either recombinant VP1u labelled with anti-FLAG antibody or anti-Gb4 antibody, washed, fixed and stained with secondary antibodies for confocal microscopy. B19V uptake (1h at 37˚C) was detected with the antibody 860-55D against intact capsids, while progeny virus (72h at 37˚C) was detected with the antibody 3113-81C against viral capsid proteins. (B) B19V uptake (1h) and infection (72h) in cells pre-incubated with recombinant VP1u (VP1u ΔC126) (S1 Fig) for 30 min at 4˚C to block the VP1uR. (C) Attachment (1h at 4˚C) and uptake (1h at 37˚C) of fluorescently labelled (Atto 488) MS2 bacteriophage capsids with conjugated recombinant VP1u ΔC126 from B19V (MS2-Atto$^{488}$-VP1u) (S1 Fig). As a control, unconjugated fluorescently labelled MS2 bacteriophage capsids were used (MS2-Atto$^{488}$). Nuclei were stained with DAPI. (D) NS1 mRNA quantification by RT-qPCR after transfection of genomic ssDNA (1h and 48h) in WT and Gb4 KO cells. The results are presented as the mean ± SD of three independent experiments. ns, not significant. (E) Detection of capsid protein expression (3113-81C; green) after transfection of genomic ssDNA (48h) in WT and Gb4 KO cells.

were able to bind and internalize into the cells without detectable differences (Fig 1C). This result indicates that virus uptake is activated by the interaction of VP1u with VP1uR, without the contribution of other capsid regions.

To better define the step of the infection where Gb4 is required, the viral ssDNA genome was extracted from native virions and directly introduced into the cells by transfection. This approach allows bypassing the cytoplasmic trafficking steps but not the second-strand synthesis, which is the first step in viral DNA replication. As shown in Fig 1D, a similar amount of

NS1 mRNAs was detected in transfected WT and Gb4 KO cells. The capacity of B19V to infect Gb4 KO cells following transfection was further confirmed by immunofluorescence microscopy with an antibody against viral capsid proteins (Fig 1E). These results together indicate that Gb4 is not required for virus attachment and uptake, but instead plays an essential role at an intracellular trafficking step before the delivery of the viral genome into the nucleus for replication.

## In the absence of Gb4, internalized B19V is arrested in the endosomal compartment

Following interaction with VP1uR, B19V is internalized by clathrin-mediated endocytosis and rapidly spreads throughout the early-late endosomes and lysosomes [32,40]. Thereafter, capsids move progressively from the endo-lysosomal compartment, which typically appears as a dense perinuclear signal, to a more dispersed spatial arrangement with limited or no co-localization with endo-lysosomal markers [40]. To further circumscribe the trafficking step where Gb4 is required, we followed the intracellular progression of the virus in the presence (WT cells) or absence of Gb4 (Gb4 KO cells) by immunofluorescence microscopy with antibodies targeting B19V capsids (860-55D) and endo-lysosomal markers (M6PR and Lamp1). The results revealed a striking difference depending on the presence or absence of Gb4. As expected, at 30 min post-infection (pi), B19V co-localized with late endosomes and lysosomes in WT and Gb4 KO cells. At 3h pi, co-localization of incoming viruses with endo-lysosomal markers decreased in WT cells and their clustered spatial distribution changed to a more scattered arrangement. In contrast, in cells lacking Gb4, the internalized virus did not show the same progression and remained associated with the endo-lysosomal markers, showing no cytoplasmic spreading of capsids as observed in Gb4-expressing cells (Fig 2A). The distinct endocytic progression of incoming capsids in presence or absence of Gb4 was further confirmed by quantitative analysis of intracellular fluorescent foci per cell (Fig 2B and S3 Fig).

This observation suggests a role of the GSL in the infectious endocytic trafficking of B19V. To further corroborate this assumption, we compared the endocytic trafficking of native B19V and MS2-VP1u particles. Similar to B19V, MS2-VP1u can internalize into the host cell through VP1uR interaction, however, these artificial capsids lack potential interaction sites harbored in the B19 capsid structure, such as a putative Gb4 binding site. Besides, MS2-VP1u lacks the PLA$_2$ domain, which is required for endosomal escape [16,41] (S1A Fig). Accordingly, following uptake these particles remained steadily associated with endo-lysosomal markers (S4 Fig). WT and Gb4 KO cells were infected with B19V for 30 min at 37˚C to allow virus internalization followed by incubation with MS2-VP1u for an additional 1h at 37˚C. At 3h pi, the cells were examined by confocal immunofluorescence microscopy. In WT cells, B19V and MS2-VP1u had a different distribution with limited co-localization. B19V appeared more scattered, while MS2-VP1u exhibited the characteristic endo-lysosomal distribution. In cells lacking Gb4, both B19V and MS2-VP1u displayed the same intracellular distribution with a strong co-localization (Fig 2C). The scattered signal observed at 3h pi in WT cells did not co-localize with early-late endosomes (EEA1, M6PR), lysosomes (Lamp1), recycling endosomes (Rab11), Golgi apparatus (Giantin) or trans-Golgi network (TGN46), however, it co-localized partially with the cis-Golgi marker GM130 (Fig 2D).

## B19V interacts with soluble and membrane-associated Gb4 in a pH-dependent manner

Various studies conducted to investigate the binding of B19V with Gb4 generated contradictory results, which was attributed to the rigorous conditions required for the interaction

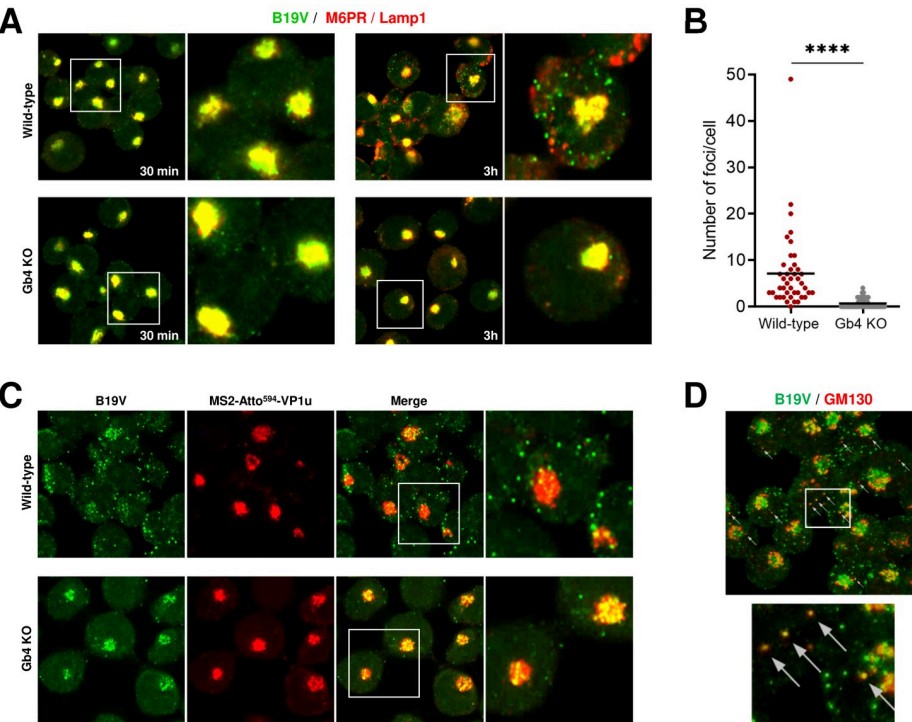

**Fig 2. In the absence of Gb4, internalized B19V is arrested in the endosomal compartment.** (A) Co-localization of incoming B19V capsids (860-55D; green) with endo-lysosomal markers (M6PR and Lamp1; red) in UT7/Epo WT and Gb4 KO cells at 30 min and 3h pi. (B) Quantitative analysis of scattered cytoplasmic foci (B19V capsid signal) not colocalizing with endocytic markers at 3h pi (S3 Fig). Unpaired Students t-test with Welch's correction (not assuming same SD) was used for statistical comparison. ****, $p<0.0001$. (C) Co-localization of incoming B19V capsids (860-55D; green) with MS2-Atto$^{594}$-VP1u in UT7/Epo WT and Gb4 KO cells at 3h pi. (D) Co-localization of B19V capsids (860-55D; green) with a cis-Golgi marker (GM130; red) in WT cells at 3h pi.

[23,25,38]. The endosomal retention of incoming capsids in cells lacking Gb4 suggests a possible interaction inside acidic endosomes, which are engaged by both the virus and the GSL shortly after internalization [40,42]. To test this hypothesis, the interaction with Gb4 expressed on human red blood cells (RBCs) was examined under characteristic acidic endosomal conditions. Interestingly, hemagglutination by B19V is routinely performed at low pH because it enhances the reaction [24,43], however, the underlying mechanism has not been elucidated. Inspired by this phenomenon, the hemagglutinating activity and the binding capacity of the virus to RBCs were quantitatively analyzed under a wide range of pH conditions. As shown in Fig 3A, hemagglutination of RBCs by B19V occurs exclusively under acidic conditions (pH < 6.4). In contrast, hemagglutination with equal amounts of minute virus of mice (MVM), a parvovirus that binds sialic acid on erythrocytes, is not influenced by pH. Increasing the number of B19 virions did not influence the results (Fig 3B). Accordingly, the low pH-dependent hemagglutination of RBCs by B19V is due to the intrinsic nature of the interaction between the virus and Gb4. As a control, the visualization of RBCs at neutral or acidic conditions by scanning electron microscopy did not reveal detectable differences in cell integrity (S5 Fig).

To confirm that the hemagglutination is caused by the interaction of the virus with Gb4, the binding of B19V to RBCs was quantified in the presence of soluble Gb4. Globotriaosylceramide (Gb3), which is the precursor of Gb4 was used as a control. Compared to pH 7.4, a sharp increase (> 2 log10) in the binding of B19V to RBCs was observed at pH 6.3. While the

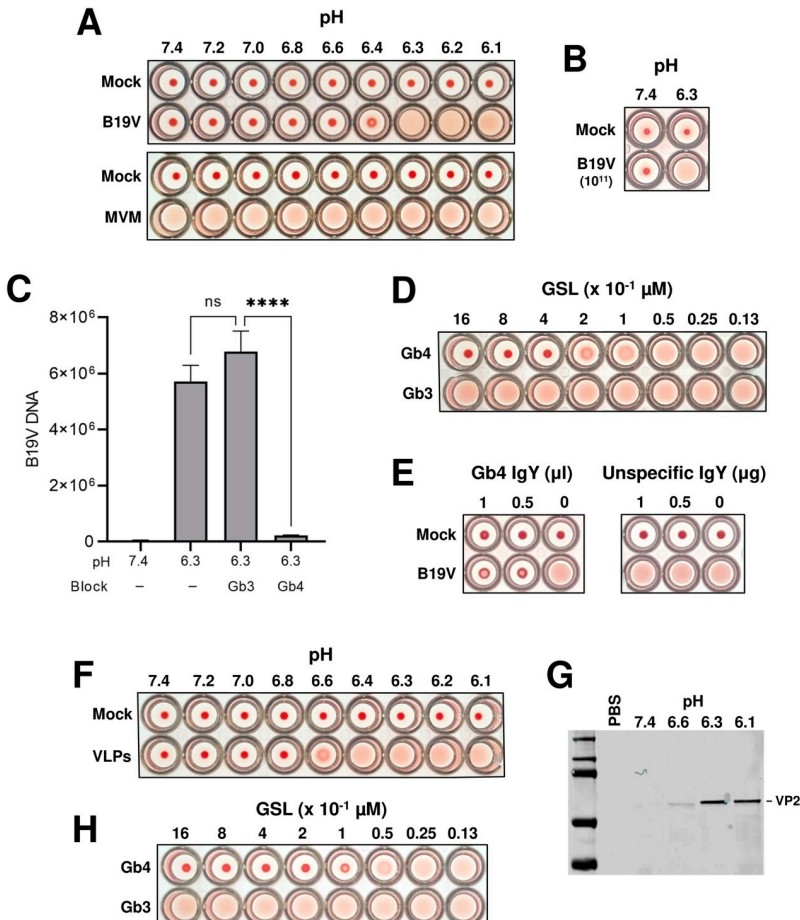

**Fig 3. B19V interacts with soluble and membrane Gb4 in a pH-dependent manner.** (A) Hemagglutination of human RBCs (0.5% in 100 μl PiBS) by B19V or MVM (5x10$^9$) at different pH values. (B) Hemagglutination of RBCs with a 20-fold increase of B19V particles at pH 7.4 and 6.3. (C) Quantitative analysis of B19V binding to RBCs at pH 7.4 and 6.3 in the presence or absence of GSLs (Gb3 or Gb4). Virus (5x10$^9$) was incubated directly with RBCs (0.5% in 100 μl PiBS) at the indicated pH or pre-blocked with 5x10$^{14}$ molecules (Gb3 or Gb4) for 1h at pH 6.3 prior to incubation with RBCs. After 1h at room temperature, the erythrocytes were washed four times with the corresponding buffer and the viral DNA was extracted and quantified. The results are presented as the mean ± SD of three independent experiments. ****, $p<0.0001$; ns, not significant. (D) Hemagglutination inhibition test in the presence or absence of different amount (0.013–1.6 μM) of GSLs (Gb3 or Gb4) at pH 6.3. (E) Hemagglutination inhibition test in the presence of anti-Gb4 or non-specific IgY antibodies at pH 6.3 (antibodies were incubated with RBCs 1h before adding the virus). (F) Hemagglutination of RBCs (0.5% in 100 μl PiBS) by VLPs (5x10$^9$) at different pH values. (G) Detection of VLPs bound to RBCs at different pH values by Western blot using antibody 3113-81C. (H) Hemagglutination inhibition test in the presence or absence of different amount (0.013–1.6 μM) of Gb3 or Gb4 at pH 6.3.

presence of soluble Gb3 did not prevent the interaction, a significant inhibition of the binding was observed in the presence of soluble Gb4 (Fig 3C). Confirming this observation, Gb4 ($>$ 0.2–0.4 μM) but not Gb3 inhibited hemagglutination of RBCs in a dose-dependent manner (Fig 3D). Likewise, hemagglutination was inhibited in the presence of a specific antibody against Gb4 (Fig 3E).

To verify that binding to Gb4 is exclusively due to a pH-mediated capsid rearrangement, the interaction was also tested with B19 virus-like particles (VLPs) consisting of VP2 (VP2-only particles). Recombinant baculoviruses that express VP2-only VLPs were prepared using the Bac-to-Bac system. The purity of the VP2 particles was verified by SDS-PAGE and

their integrity was examined by electron microscopy and by dot blot with an antibody against intact capsids (S6 Fig). Similar to the plasma-derived B19V, hemagglutination of erythrocytes by VLPs was also pH-dependent (Fig 3F). A minor shift in the pH required for hemagglutination between VLPs (6.4 full, 6.6 partial) and B19V (6.3 full, 6.4 partial) was observed. Since the accurate quantification of B19V by PCR is not possible for VLPs, this minor variation might be explained by differences in the number of particles applied. The pH-dependent binding of VLPs to erythrocytes was examined by Western blot with an antibody against VP2. The binding at the different pH values correlated well with the HA (Fig 3G). Similar to native B19V, hemagglutination by VLPs was inhibited in a dose-dependent manner by Gb4 but not by Gb3 (Fig 3H). Together, these results demonstrate that B19V does not bind to Gb4 expressed on cellular membranes, which are typically exposed to neutral pH conditions. The interaction occurs exclusively under acidic conditions and is mediated by the VP2 region of the capsid.

## Different to the extracellular environment, the early endosomal compartment provides optimal conditions for Gb4 interaction

B19V binding to Gb4 expressed on human erythrocytes was tested at various pH conditions and quantified by qPCR. The results revealed a progressive increase in affinity at decreasing pH values, reaching a maximum at pH 6.0 (Fig 4A), which corresponds to the average pH measured in early endosomes [44,45]. At neutral pH, the binding affinity decreased more than 3 logs to background levels, confirming that B19V does not recognize Gb4 expressed on the plasma membrane.

Parvovirus capsid proteins are fine-tuned to rearrange in response to pH conditions, but also to other cellular cues encountered during entry. Besides the low pH, early endosomes are characterized by a depleted $Ca^{2+}$ environment [46,47]. Divalent cations have been shown to play important roles in the capsid stability of parvoviruses and their depletion can trigger structural rearrangements and alter capsid integrity [48–50]. The influence of divalent cations in the interaction was examined at pH 6.3, which is close to the hemagglutination threshold and therefore more sensitive to affinity fluctuations between the virus and the GSL. Neither the sequestration of $Ca^{2+}$ by EGTA or $Mg^{2+}$ by EDTA nor their addition influenced the low pH-dependent hemagglutination activity of B19V. The HA was also undisturbed in a complex solution, such as cell culture media (MEM, minimal essential medium) (Fig 4B). Low pH-

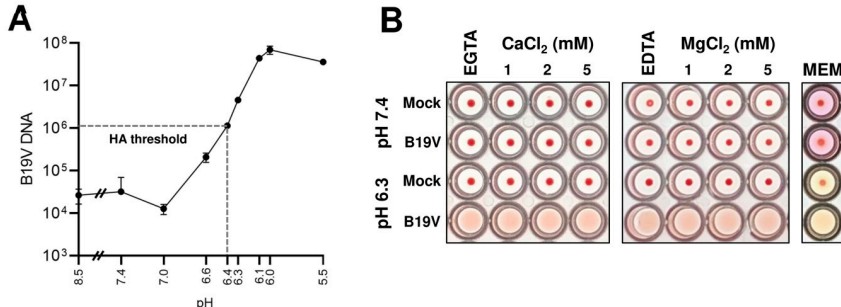

**Fig 4. Determination of the optimal conditions required for B19V and Gb4 interaction.** (A) Determination of the optimal pH for B19V binding to Gb4. RBCs (0.5% in 100 µl PiBS) were incubated with B19V (5x10⁹) at pH values ranging from 8.5 to 5.5. After 1h, cells were washed in the corresponding buffer, and viral DNA was extracted and quantified by qPCR. (B) Hemagglutination at neutral (7.4) or acidic (6.3) pH was carried out in the presence of divalent cations ($Ca^{2+}$ or $Mg^{2+}$), chelating agents (5 mM EGTA or EDTA) or in minimal essential medium (MEM). B19V was incubated with the different buffers for 1h before incubation with the erythrocytes. HA, hemagglutination assay.

mediated binding of B19V to Gb4 was also independent of the temperature (S7 Fig). These results indicate that contrary to the extracellular milieu, the conditions found in the early endosomal compartment are optimal for the interaction between incoming viruses and Gb4.

## pH acts as a switch to regulate the affinity between B19V and Gb4

The interaction of viruses with cellular partners is highly dynamic and influenced by the different environmental conditions encountered during the process of entry, resulting in affinity fluctuations. These affinity changes are finely tuned to promote binding or dissociation with the target molecules. During entry, B19V is initially exposed to the acidic conditions of the endosomal compartment, followed by the neutral pH of the cytosol, where the dissociation from the GSL would facilitate the virus targeting to the nucleus. To investigate pH-dependent changes in binding affinity, the dissociation of B19V from Gb4 expressed on human erythrocytes was examined by adjusting the pH from 6.3 to 7.4. As shown in Fig 5A, more than 95% of the particles bound to Gb4 on erythrocyte membranes at pH 6.3 dissociated when the pH was restored to 7.4. Moreover, preincubation of viruses or RBCs separately at pH 6.3 did not allow hemagglutination at neutral pH (S8 Fig). Additionally, the reversibility of the interaction was tested by direct visualization of the hemagglutination reaction with VLPs. Under light microscopy, the erythrocytes appeared dispersed at pH 7.4 and formed large clusters at pH 6.3, conforming with the HA. The erythrocyte clusters were not detected when the acidic pH was neutralized (Fig 5B), indicating that binding of VLPs to Gb4 is also reversible.

Although the interaction is reversible, binding of B19V to Gb4 may trigger irreversible capsid conformational changes that prepare the virus for subsequent infection steps or even render the incoming capsids independent of Gb4. To test this hypothesis, viruses bound to Gb4 at low pH were released by exposure to neutral pH and used to infect WT and Gb4 KO cells. The previous interaction of the virus with Gb4 on the surface of RBCs at low pH changed neither their capacity to infect WT cells nor their inability to infect Gb4 KO cells (Fig 5C). These results together reveal that pH acts as a regulatory switch, modulating the affinity between the virus and the GSL.

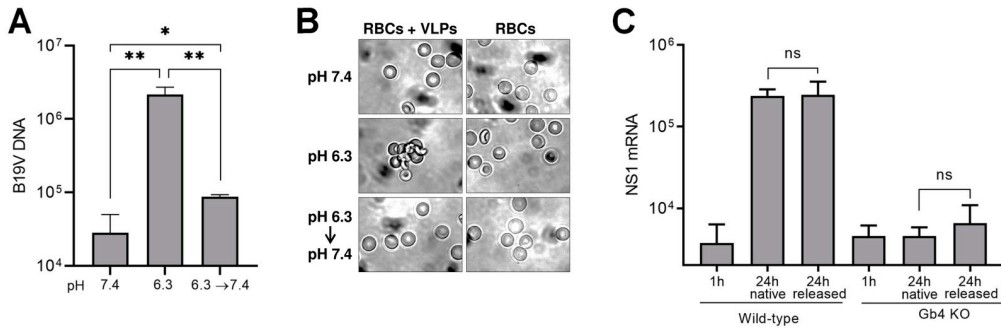

**Fig 5. pH acts as an affinity switch to regulate binding and dissociation between B19V and Gb4.** (A) B19V ($5 \cdot 10^9$) was incubated with RBCs (0.5% in 100 μl PiBS) for 1h at pH 7.4 or 6.3. The cell suspensions were washed twice with a buffer of the same pH, except for one sample incubated at pH 6.3 and washed at pH 7.4. RBCs were incubated 30 additional minutes in the washing buffer at room temperature, washed twice, and viral DNA was extracted and quantified by qPCR. The results are presented as the mean ± SD of three independent experiments. *, $p < 0.05$; **, $p < 0.01$. (B) Visualization of the hemagglutination reaction by VLPs. RBCs were incubated with VLPs ($5 \cdot 10^9$) at pH 7.4 or 6.3 for 1h at room temperature. Prior to visualization by phase contrast microscopy, the samples were diluted with a buffer of the same pH or neutralized to pH 7.4. (C) Infectivity of Gb4-dissociated virus. Viruses bound to RBCs at pH 6.3 and dissociated at pH 7.4 were quantified by qPCR. Equal amounts of native and Gb4-dissociated virus were added to UT7/Epo WT or Gb4 KO cells. After 1h or 24h, cells were washed four times and NS1 mRNA was extracted and quantified by RT-qPCR. The results are presented as the mean ± SD of three independent experiments. ns, not significant.

## Low pH-mediated interaction of B19V with Gb4 initiates active membrane processes

Multivalent interactions of certain ligands, such as toxins, lectins and viruses with GSLs have been shown to change membrane properties resulting in membrane invaginations and vesicle formation [51–58]. The multimeric configuration of the B19V capsid and the small size of Gb4 would favor multivalent interactions with the GSL and the subsequent changes in membrane properties. To examine the capacity of B19V binding to Gb4 to initiate active membrane processes, we sought to induce the interaction of the virus with Gb4 expressed on UT7/Epo cells. Even though Gb4 is expressed abundantly on the plasma membrane of UT7/Epo cells, the neutral pH conditions at the cell surface prohibit the interaction. To force the binding to the GSL, WT and Gb4 KO cells were incubated with the virus at pH 6.3 for 1h at 37˚C. In WT cells, a 92-fold average increase in virus attachment was observed at acidic compared to neutral pH. In sharp contrast, the pH conditions had no significant effect on virus attachment in Gb4 KO cells, confirming that Gb4 is responsible for the pH-dependent binding enhancement in WT cells (Fig 6A). Virus attachment was also analyzed under VP1uR blocking conditions. To this end, WT and KO cells were either incubated with functional (ΔC126) or non-functional (ΔN29, lacking the RBD) recombinant VP1u (S1A Fig). Subsequently, the cells were incubated with B19V at 37˚C for 1h at pH 7.4 or 6.3. As expected, when VP1uR was not blocked, WT and KO cells exhibited a comparable virus binding at pH 7.4, whereas at pH 6.3 virus binding was substantially increased only in WT cells. Under VP1uR blocking conditions, virus attachment was inhibited, except for WT cells at pH 6.3, which represents viruses bound exclusively to Gb4 (Fig 6B).

We next analyzed the capacity of B19V bound to Gb4 to trigger membrane processes resulting in virus uptake. To this end, WT cells were preincubated with recombinant VP1u ΔC126 to block VP1uR. Subsequently, the virus was incubated with the cells at pH 6.3 for 1h at 4˚C to allow attachment to Gb4 or 37˚C to allow attachment and uptake. After the incubation, the cells were washed with PBS pH 7.4 to detach non-internalized capsids. While most of the viral particles were removed from cells incubated at 4˚C, cells incubated at 37˚C displayed a strong intracellular accumulation of capsids (Fig 6C).

To examine the capacity of the Gb4-mediated uptake to initiate the infection, viruses were incubated with cells under VP1uR-blocking conditions at pH 6.3 to allow binding to Gb4. Subsequently, NS1 mRNA was quantified after 24h by qPCR and the presence of progeny capsids was examined after 72h by immunofluorescence. Although there was a modest increase of NS1 mRNA mediated by Gb4 entry (Fig 6D), no capsid proteins were produced after three days (Fig 6E). Expectedly, Gb4-mediated uptake did not result in infection, since the interaction with extracellular Gb4 was induced under conditions that are not expected to occur during the natural infection. However, this experimental approach demonstrated that binding of B19V to Gb4 can stimulate active membrane processes similar to those observed with other ligands interacting with GSLs.

VLPs consisting of VP2, lack the entire VP1. Without the VP1u, these particles are unable to recognize the VP1uR and thus cannot be internalized into permissive cells [32,36]. These capsids are particularly useful because they allow the study of Gb4 interaction in a more specific way without the interference of the VP1uR.

The pH-dependent interaction of VLPs with WT and Gb4 KO UT7/Epo cells was examined by Western blot with an antibody against VP2. The results showed a low pH-dependent interaction with Gb4 exclusively in WT cells (Fig 7A). This result was corroborated by immunofluorescence microscopy with an antibody against intact capsids (Fig 7B).

We next tested the capacity of VLPs bound to Gb4 to internalize into UT7/Epo cells. Cells were incubated with VLPs at pH 6.3 at 4˚C or 37˚C. After 1h, the cells were washed, fixed and

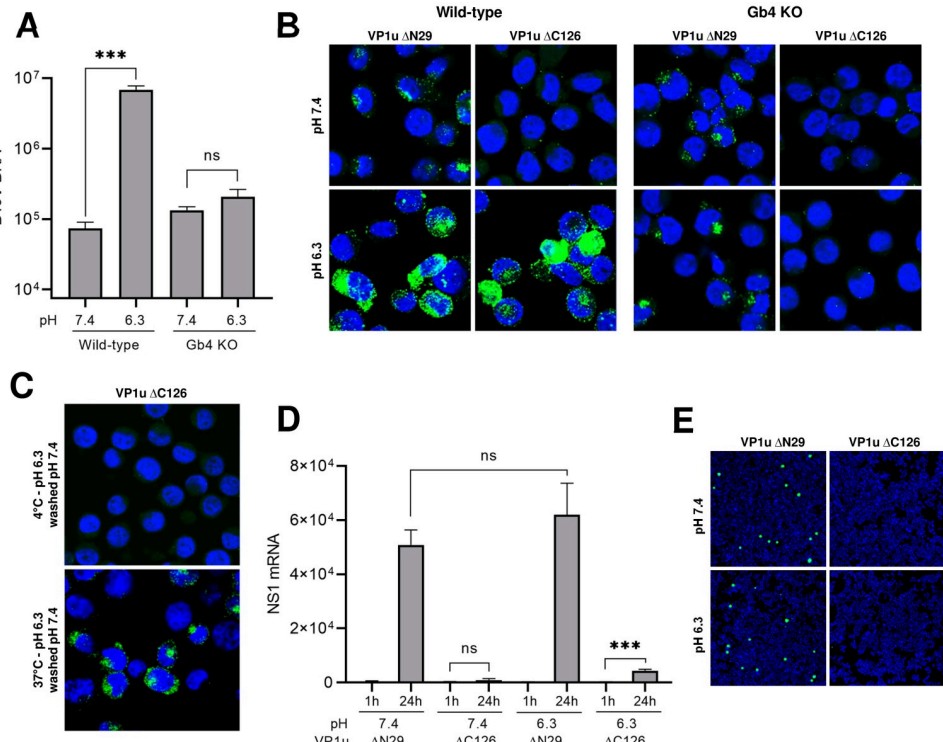

**Fig 6. Low pH-mediated interaction of B19V with Gb4 initiates active membrane processes.** (A) Quantification of B19V attachment to UT7/Epo WT and Gb4 KO cells at neutral and acidic pH. Cells were infected with B19V ($10^4$ geq/cell) at 37°C for 1h followed by four washes. DNA was extracted and quantified by qPCR. The results are presented as the mean ± SD of three independent experiments. ***$p < 0.001$; ns, not significant. (B) Detection of B19V capsids (860-55D) in UT7/Epo WT and Gb4 KO cells under neutral or acidic pH by IF. Cells were incubated for 30 min with functional (ΔC126) to block the VP1uR or non-functional (ΔN29) recombinant VP1u, as a control (S1 Fig) at 4°C. Subsequently, B19V ($510^4$ geq/cell) was added for 1h at 37°C. (C) Gb4-mediated uptake of native B19V. Cells were preincubated with functional VP1u ΔC126 for 1h at 4°C prior to infection to block the VP1uR followed by incubation with B19V at 4°C or 37°C for 1h at pH 6.3. Non-internalized virus was removed by a washing step at neutral pH. Internalized viruses were detected by IF with antibody 860-55D against capsids. (D) Infectivity assay at neutral and acidic pH. WT cells were incubated for 30 min at 4°C with functional recombinant VP1u (ΔC126) to block the VP1uR or non-functional (ΔN29), as a control. Subsequently, B19V ($510^4$ geq/cell) was added for 1h at 37°C in a buffer with the indicated pH. Cells were washed after 1h or further incubated for 24h at 37°C. NS1 mRNA was extracted and quantified by RT-qPCR. The results are presented as the mean ± SD of three independent experiments. ns, not significant. (E) Alternatively, 72h post-infection, capsid protein expression was examined by IF with antibody 3113-81C.

examined by confocal immunofluorescence microscopy. Pictures taken from the top and the middle sections revealed the presence of internalized capsids in cells incubated at 37°C, but not at 4°C (Fig 7C). Alternatively, following 1h incubation at 4°C or 37°C, cells were washed with PBS pH 7.4 to detach non-internalized capsids. While most of the viral particles were removed from cells incubated at 4°C, cells incubated at 37°C displayed a strong intracellular signal (Fig 7D). Collectively, these results confirm that the low pH-mediated interaction of B19V with Gb4 is mediated by the VP2 region and induces changes in membrane properties resulting in vesicle formation.

## Erythrocytes do not play a significant role as viral decoy targets during B19V infection

B19V can establish high-titer viremia during the acute phase of the infection. Gb4 is the most abundant neutral GSL expressed on RBCs [29,30]. Since RBCs cannot be infected by viruses,

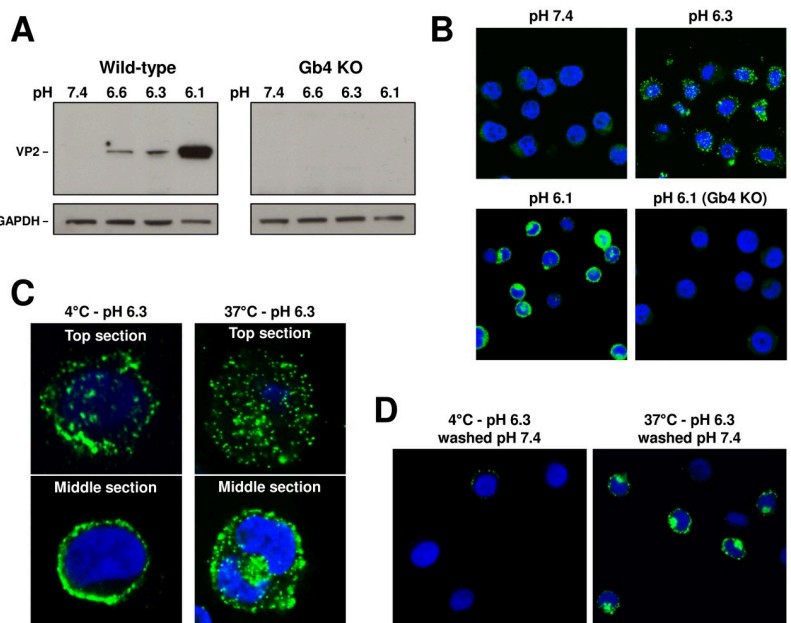

**Fig 7. pH-mediated interaction of VP2-only particles with Gb4 triggers the uptake process independently of the VP1uR.** (A) Detection of VLPs bound to WT and Gb4 KO UT7/Epo cells at different pH values by Western blot using antibody 3113-81C. VLPs ($10^{10}$) were incubated with cells ($3 \times 10^5$) at the indicated pH values for 1h at 37°C. GAPDH expression was used as a loading control. (B) Detection of VLPs (860-55D) in UT7/Epo WT and Gb4 KO cells under neutral or acidic pH by IF. (C) Gb4-mediated uptake of VLPs. Top and middle sections of cells incubated with VLPs at 4°C or 37°C for 1h at pH 6.3. (D) Gb4-mediated uptake of VLPs. Cells were incubated with VLPs at 4°C or 37°C for 1h at pH 6.3. Non-internalized virus was removed by a washing step at neutral pH. Internalized particles were detected by IF with antibody 860-55D against capsids.

binding of B19V to Gb4 expressed on erythrocytes would trap the virus and hinder the infection. The interaction of B19V with RBCs was tested in the natural environment of the blood. To this end, B19V was spiked into fresh blood samples (pH 7.4) without B19V-specific antibodies. After incubation for 1h, the erythrocytes were separated from the plasma by centrifugation and washed at neutral pH. As shown in Fig 8A, B19V was consistently found in the plasma fraction in three distinct blood samples. The absence of significant B19V binding to RBCs was further confirmed by immunofluorescence microscopy with an antibody against intact capsids. As expected, decreasing the pH to 6.3 triggered a substantial binding of the virus to RBCs (Fig 8B). Besides the neutral pH of the blood, an unknown component(s) of the plasma has been shown to inhibit the hemagglutination activity of B19V [24]. Consistent with this observation, the binding efficiency of B19V to RBCs increased approximately 10-fold when no blood plasma components were present (Fig 8C). Since the conditions for the interaction with Gb4 are largely suboptimal in the blood, we conclude that despite the abundant expression of Gb4, the erythrocytes do not play a significant role as viral decoy targets during B19V viremia.

## Discussion

B19V has a marked tropism for erythroid progenitor cells (EPCs) in the bone marrow [17–19]. The narrow tropism of B19V is mediated by multiple factors highly restricted to the Epo-dependent erythroid differentiation stages [21,59–64]. A virus demanding such strict conditions for replication would also require a selective receptor exclusively expressed in the target cells. This strategy would allow the virus to avoid nonpermissive cells, which have the potential

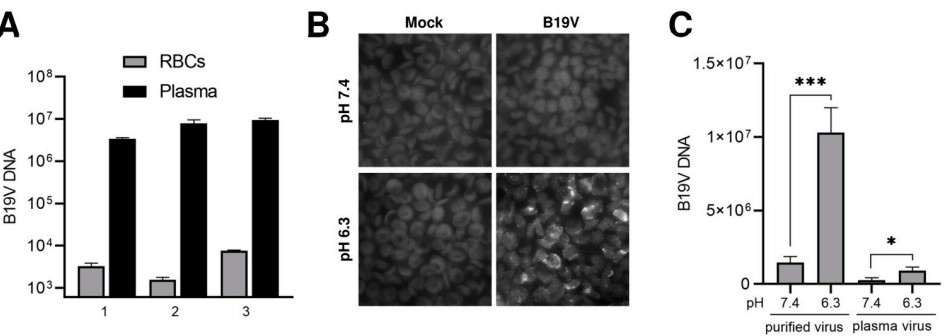

**Fig 8. Erythrocytes do not play a significant role as viral decoy targets during B19V viremia.** (A) Freshly collected blood samples (with EDTA as anticoagulant) and tested negative for antibodies against B19V, were spiked with B19V ($10^9$ virions) and incubated for 1h at 37°C. The plasma and the RBC fractions were separated by centrifugation, the RBCs were washed with PBS (pH 7.4) and the viral DNA was extracted from both fractions and quantified by qPCR. (B) Detection of B19V in the RBC fraction by IF with an antibody against intact capsids (860-55D). (C) A component (s) in human plasma inhibits binding of B19V to RBCs. RBCs (0.5% in 100 µl PiBS) were incubated at pH 7.4 or 6.3 with B19V ($5x10^9$) directly from an infected plasma sample or after purification by iodixanol density gradient centrifugation. After 1h at room temperature, the erythrocytes were washed four times with the corresponding buffer and viral DNA was extracted and quantified. The results are presented as the mean ± SD of three independent experiments. *, $p<0.05$; ***, $p<0.001$.

to hamper the infection by diverting viruses away from target tissues. The neutral glycosphingolipid (GSL) globoside (Gb4) has been historically considered the primary cellular receptor of B19V [23,26]. However, Gb4 is not the receptor that would be expected for a virus with the remarkable restricted tropism of B19V. Gb4 is expressed in a multitude of cell types that are not permissive to the infection [27,28], and is the major neutral GSL of RBCs, which cannot support viral infections [29,30]. Accordingly, the interaction between the virus and the GSL either does not occur or requires strict conditions.

In an earlier study, we identified a functional RBD in the VP1u region of B19V, which mediates virus attachment and uptake in EPCs through interaction with a yet unidentified receptor, referred to as VP1uR [11,32]. In contrast to the ubiquitous expression of Gb4, VP1uR expression is restricted to EPCs, which are the only cells that B19V can productively infect [19,32]. In a follow-up study, we showed that B19V was able to attach and internalize cells expressing VP1uR but lacking Gb4. However, in the absence of Gb4, the internalized virus was unable to initiate the infection [37]. These findings together reveal that VP1uR is the cellular receptor of B19V responsible for the restricted virus uptake in permissive erythroid cells, whereas Gb4 is an essential factor required at a post-internalization step.

In this study, the mechanism of the interaction of B19V with Gb4 and the step of the infection where the GSL is required were investigated. We confirmed that the genetic removal of the gene encoding for Gb4 in UT7/Epo cells does not disturb the expression and function of the VP1uR, and consequently, the uptake of B19V is also not affected (Fig 1A–1C). However, Gb4 is required after virus internalization and before the delivery of the ssDNA into the nucleus for replication (Fig 1D and 1E). It has been previously shown that during the first hour following B19V uptake in UT7/Epo cells, the incoming capsids colocalize with endo-lysosomal markers, appearing as a dense perinuclear signal. Subsequently, the virus signal becomes more scattered and colocalization with endocytic markers decreases gradually [40]. As shown in Fig 2A, 30 min post-internalization in WT and Gb4 KO cells, the virus signal colocalizes intensively with endo-lysosomal markers. As expected, 3h pi in WT cells, capsid signal becomes more scattered and less co-localized with endo-lysosomal markers. In contrast, in cells lacking Gb4, the virus signal does not progress and remains associated with endosomal

markers, suggesting a role of Gb4 in the infectious endocytic trafficking of B19V. Furthermore, in Gb4 KO cells, B19V behaves like artificial MS2-VP1u capsids, which are unable to escape from endosomes (Fig 2B and S3 Fig).

The finding that pH acts as a switch to modulate the affinity between B19V and Gb4 has major implications in the virus tropism, infection and spread. Under neutral conditions, which are characteristic of the extracellular milieu, B19V does not interact with the ubiquitously expressed Gb4. This strategy prevents the redirection of the virus to nonpermissive tissues facilitating the selective targeting of the EPCs in the bone marrow. We and others observed a strong association of B19V to RBCs in viremic blood samples [35,65]. The blood sample used in our studies was acidic due to anaerobic glycolysis, a process that occurs naturally during blood storage [66]. In the study of Chehadeh et al., the RBCs were washed in acidic PBS and stored in Alsever's solution, which is an acidic solution (pH 6.1) routinely used as an anticoagulant and preservative. In other studies, when B19V was spiked into a fresh blood sample to mimic viremia, the virus was mostly associated with the plasma fraction and those found in the RBC fraction were easily removed by a washing step. [67]. This result was confirmed in our studies, where no significant binding to RBCs was observed when B19V was spiked into fresh blood samples with an experimentally verified neutral pH (Fig 8A and 8B). Besides a suboptimal pH for Gb4 binding, blood plasma contains an inhibitor(s) that interferes with hemagglutination (Fig 8C) [24]. Our results revealed that during B19V viremia, the fairly constant neutral pH of the blood and the presence of an inhibitor(s) hinder the stable binding of the virus to Gb4 on RBCs, which would otherwise divert the virus with an overwhelming amount of decoy targets and thereby hamper the infection.

The acidic pH conditions required for the interaction exclude the binding to Gb4 expressed on the plasma membrane, which is typically exposed to neutral pH conditions. However, Gb4 is also found in intracellular compartments. GSLs are continuously internalized from the cell surface by clathrin-dependent and independent mechanisms in invaginated vesicles. These vesicles fuse with early endosomes, resulting in the glycan component of the GSL facing the vesicle lumen [42,68]. From early endosomes, GSLs can be recycled back to the plasma membrane, transported to the Golgi apparatus or to late endosomes and finally lysosomes where they undergo terminal degradation by specific lysosomal enzymes [42,69–71]. Although B19V is internalized via clathrin-dependent endocytosis [40], and Gb4 is mostly internalized via clathrin-independent endocytosis [42], both reach the early endosomes. The acidic luminal pH in early endosomes around 6.0 and the depleted $Ca^{2+}$ levels coincide with the optimal conditions required for the interaction with Gb4 (Fig 4). Accordingly, incoming B19V can potentially interact with Gb4 inside early endosomes as an essential step in the infectious trafficking. In line with this assumption, the absence of Gb4 resulted in endosomal retention of the incoming capsids (Fig 2).

Multivalent interactions of certain ligands, such as bacterial toxins, lectins and viruses, with GSLs trigger membrane curvature and invaginations that ultimately result in vesicle formation [51–58]. In our studies, the interaction under acidic conditions of B19V and VLPs with Gb4 in the exoplasmic membrane leaflet resulted in virus uptake (Figs 6C, 7C and 7D), suggesting that similar active membrane processes are induced by the multivalent binding of the capsid with Gb4 molecules. Further research will aim to characterize changes in membrane properties mediated by the interaction of the virus with GSL-enriched areas of the endosomal membrane and the influence of additional intracellular factors in the interplay between B19V and Gb4.

In summary, this study provides mechanistic insight into the interaction of B19V with Gb4 and its essential role as an intracellular interacting partner required for infectious trafficking. The finding that pH acts as an affinity switch to modulate the interaction between the virus and the GSL contributes to a better understanding of B19V restricted tropism, infection and

spread. In the future, studies will aim at elucidating the precise function of Gb4 in B19V endocytic trafficking, which will deepen our understanding of membrane dynamics induced by the interaction of viruses with GSLs and inspire novel strategies interfering with the early steps of the infection.

## Materials and methods

### Cells and viruses

The human megakaryoblastoid cells UT7/Epo were cultured in Eagle's minimal essential medium (MEM) containing 5% fetal calf serum (FCS) along with 2 U/ml recombinant erythropoietin (Epo). Whole blood samples from anonymous donors were washed three times with PBS. Packed red blood cells (RBCs) were resuspended in an equal volume of Alsever's solution (4.2 g/l NaCl, 8 g/l sodium citrate, 0.55 g/l citric acid, 20.5 g/l dextrose) and stored at 4˚C. ExpiSf9 cells for recombinant baculovirus production were cultured at 27˚C and 125 rpm in ExpiSf CD Medium (Thermo Fisher Scientific, Waltham, MA). B19V-infected plasma sample was obtained from a donation center (CSL Behring AG, Charlotte, NC) and virus concentration ($3x10^9$ geq/μl) was determined by qPCR. B19V was concentrated by ultracentrifugation through a 20% sucrose cushion and further purified by iodixanol density gradient ultracentrifugation, as previously described [50]. Plaque-purified MVM was obtained from ATCC (VR-1346).

### Antibodies

The human monoclonal antibody 860-55D against intact capsids was purchased from Mikrogen (Neuried, Germany). The monoclonal mouse antibody 3113-81C (US Biological, Boston, MA) was used for the detection of viral proteins by Western blot as well as for the detection of progeny virus by immunofluorescence. A polyclonal chicken anti-Gb4 IgY antibody was a gift from J. Müthing (University of Münster). Antibodies against late endosomes (M6PR, ab2733), lysosomes (Lamp1, ab25630), cis-Golgi (GM130, ab52649) and GAPDH (ab9485) were obtained from abcam (Cambridge, UK). A rat anti-FLAG monoclonal antibody (200474) was purchased from Agilent (Santa Clara, CA). MS2 capsid proteins were detected with a polyclonal rabbit antibody (ABE76-I, Merck Millipore, France).

### Generation of MS2-VP1u bioconjugate

Fluorescent MS2 VLPs bioconjugated to B19V VP1u were produced as previously described [19]. Briefly, MS2 coat protein or truncated VP1u (ΔC126/ΔN29) proteins were expressed in *E. coli* BL21(DE3) cells for 4h at 37˚C. Assembled MS2 capsids in cell lysate were purified by ultracentrifugation through a sucrose cushion. Recombinant VP1u was purified twice with nickel nitrilotriacetic acid (Ni-NTA) agarose. Chemical crosslinking between MS2 VLPs and truncated VP1u proteins was carried out in two steps. First, surface lysines on MS2 capsids were modified with fluorescent dyes (NHS-Atto488 or NHS-Atto633) and the heterobifunctional maleimide-PEG24-N-hydroxysuccinimide ester crosslinker (Thermo Fisher Scientific). Second, purified fluorescent maleimide-activated MS2 capsids were incubated with reduced VP1u proteins to achieve bioconjugation. After quenching of the reaction, MS2-VP1u constructs were pelleted by ultracentrifugation.

### Production and purification of B19 virus-like particles

Recombinant B19 virus-like particles (VLPs) consisting of VP2 were produced using the ExpiSf Expression System Starter Kit (Thermo Fisher Scientific) following the manufacturer's

instructions. Briefly, the B19 VP2 gene was cloned into a pFastBac1 plasmid and used to create recombinant bacmids, which were transfected into ExpiSf9 cells to generate recombinant baculovirus. This virus was able to express VP2 particles by infection of ExpiSf9 cells at a multiplicity of infection of 5. Infected ExpiSf9 cells were lysed 72h pi and the assembled B19 VP2 particles were purified by ultracentrifugation through a sucrose cushion followed by a separation on a sucrose gradient. Positive fractions were identified by dot blot and exchanged into a storage buffer (20 mM Tris-HCl, [pH 7.8], 10 mM NaCl, 2 mM $MgCl_2$) using desalting columns. Quantification of VP2 particles was determined by absorbance at A280 with NanoDrop (NanoDrop2000, Thermo Fisher Scientific), as well as by comparison to a reference B19V sample on a dot blot.

## Immunofluorescence

For surface staining, UT7/Epo cells were incubated at 4˚C with anti-Gb4 antibody or with anti-FLAG-tag labelled recombinant VP1u prior to fixation. UT7/Epo cells were fixed with a mixture of methanol and acetone (1:1) at -20˚C for 4 min. RBCs were fixed with 1% glutaraldehyde at room temperature for 10 min. Fixed cells were blocked with 10% goat-serum prior to staining with antibodies. Bound primary antibodies were detected with secondary antibodies with conjugated Alexa Fluor dyes and analyzed using confocal microscopy (LSM 880, Zeiss, Germany).

## Virus binding and internalization

For each experiment, either UT7/Epo cells ($3x10^5$) or alternatively a 0.5% RBC suspension were prepared in 100 μl PiBS (20 mM piperazine-N,N′-bis[2-ethanesulfonic acid], 123 mM NaCl, 2.5 mM KCl) of varying pH or PBS. B19V was added to UT7/Epo cells ($10^4$ geq/cell for PCR analysis, $5x10^4$ geq/cell for immunofluorescence analysis) or to RBCs ($5x10^9$) and incubated at 4˚C or 37˚C for 1h. Cells were washed 4 times at room temperature. Subsequently, the samples were prepared for immunofluorescence analysis or qPCR. For qPCR, B19V DNA was isolated by the DNeasy Blood & Tissue Kit (Qiagen, Hilden, Germany) according to the manufacturer's protocol. The following primers were used, forward primer: 5'-GGGGCAGCATGT GTTAA-3'; reverse: 5'- AGTGTTCCAGTATATGGCATGG-3'.

## Transfection

B19V DNA was isolated and quantified as described earlier. The viral DNA was transfected into UT7/Epo cells ($2x10^5$) seeded one day prior using lipofectamine 3000 reagent (Invitrogen) according to the manufacturer's instructions. B19V NS1 mRNA and viral capsid proteins were analyzed 2 days post infection using immunofluorescence as described above or RT-qPCR. Viral mRNA transcripts were isolated using the Dynabeads mRNA DIRECT Kit (Invitrogen) according to the proposed protocol. Identical primers as indicated above were employed.

## NS1 mRNA and capsid protein expression

Infection of UT7/Epo cells was examined by quantification of viral NS1 mRNA and by immunofluorescence staining of viral proteins. Cells were washed four times 1h after virus binding and incubated in medium for up to three days. The cells were harvested and washed four times with PBS. For immunofluorescence analysis the cells were fixed as described above and stained with a monoclonal mouse antibody against the viral capsid proteins followed by staining with a goat anti-mouse IgG Alexa Fluor 488 (Invitrogen). Viral mRNA was extracted and analyzed as described for transfected cells.

## Hemagglutination assay

RBCs were washed 3 times and brought to a concentration of 1% in PiBS. Glycosphingolipids (Gb3 [Globotriaosylceramide] and Gb4, Matreya LLC, PA) were dissolved in DMSO (5 mg/ml and 25 mg/ml respectively) and diluted to the desired concentration immediately before use. When applicable, B19V or VP2 particles ($5x10^9$) were incubated for 30 min in 50 μl of the indicated buffers along with Gb3/Gb4 prior to hemagglutination experiments. 50 μl 1% RBC solution were then added to each well and incubated for 1h at room temperature.

## Western blot analysis of VP2 particles binding

A total of $10^{10}$ VP2 particles were added to either a 0.5% RBC solution or to $3x10^5$ UT7/Epo cells in 100 μl PiBS of varying pH. The cell suspension was incubated for 1h at 37 ˚C. The cells were subsequently washed three times and resuspended in 2x Laemmli buffer containing 0.1M dithiothreitol. Samples were boiled and resolved on a 10% SDS-PAGE and transferred to a PVDF membrane. The membrane was blocked overnight at 4˚C using 5% milk in TBS-T. Viral proteins were detected using the same mouse antibody described above followed by far-red fluorescent based detection using an 680RD goat anti-Mouse IgG secondary antibody (LI-COR Biosciences, Lincoln, NE) in the case of RBCs or with chemiluminescent detection using an HRP-conjugate for the UT7/Epo cells.

## Cell cycle analysis

UT7/Epo cells ($1.5x10^5$) were seeded in 1 ml MEM in a 12-well plate and infected with B19V containing plasma (40'000 geq per cell) or mock infected. Cells were harvested 3 days post infection and washed with PBS containing 1% albumin (PBSA), resuspended in 300 μl PBSA and fixed by dropwise addition of 700 μl ethanol cooled to -20˚C. Tubes were carefully inverted five times stored at 4˚C for 1h. The cells were pelleted and washed twice with PBSA. Cells were incubated with 100 μg RNase A for 30 min at 37˚C and stained with 1 μg 4′,6-diamidino-2-phenylindole (DAPI). Cells were sorted on a Cytoflex flow cytometer (Beckman Coulter) and analyzed using FCS express 7 (De Novo Software).

## Analysis

Data analysis was performed using GraphPad Prism *and* presented as the mean of three independent experiments ± standard deviation (SD). Differences in the binding of B19V to Gb4, NS1 mRNA synthesis, and cell cycle arrest at the G2/M-phase, were evaluated by Student's t-test. A P value less than 0.05 was considered statistically significant.

## Supporting information

**S1 Fig. B19V VP1u constructs and engineered MS2 particles.** (A) Schematic depiction of the functional (ΔC126) and non-functional (ΔN29) recombinant VP1u constructs. (B) SDS-PAGE of purified recombinant VP1u constructs under reducing conditions. (C) Schematic depiction of an MS2 particle showing Atto fluorophores and VP1u constructs incorporated on the capsid surface. (D) Crosslinking between recombinant MS2 capsid proteins and VP1u constructs was verified by Western blot using an anti-MS2 antibody.
(TIF)

**S2 Fig. B19V induces cell cycle arrest in WT but not in Gb4 KO UT7/Epo cells.** Cells were fixed 3d pi and cellular DNA was stained with DAPI. Cell cycle progression was analyzed using flow cytometry. The results are presented as the mean ± SD of three independent

experiments. [**], $p < 0.01$; ns, not significant.
(TIF)

**S3 Fig. Quantitative analysis of intracellular fluorescent foci per cell.** Cells exhibiting perinuclear cluster of endosomes in focus (encircled by a dotted line) were selected for analysis. Distinct, clearly visible fluorescent spots (B19V capsids; green) not colocalizing with endocytic markers (red) were counted.
(TIF)

**S4 Fig. Internalized MS2-VP1u particles remain sequestered inside the endosomes.** UT7/Epo cells ($3x10^5$) were incubated with 2 μl Atto 488-labeled MS2-VP1u at 4°C for 1h, washed and further incubated at 37°C for 30 min and 3h. Cells were fixed and labeled with antibodies against late endosomes (M6PR) and lysosomes (Lamp1) and visualized under confocal microscopy.
(TIF)

**S5 Fig. RBCs integrity is not compromised by exposure to mild acidic conditions.** Scanning electron microscopy of RBCs exposed to pH 7.4 or 6.3 for 2h. RBCs were fixed with 1% glutaraldehyde, dehydrated by subsequent treatment with increasing concentrations of ethanol. Specimen were mounted and analyzed on a scanning electron microscope (Zeiss) with a 100'000-fold magnification. Bar, 10 μm.
(TIF)

**S6 Fig. Purity and integrity of B19 VLPs.** (A) Capsid protein purity of VLPs (VP2-only particles) was verified by SDS-PAGE. Capsid integrity was analyzed by electron microscopy (B), and by dot blot hybridization with an antibody against intact capsids (860-55D) (C). Bar; 100 μm.
(TIF)

**S7 Fig. Effect of temperature on the low pH-mediated interaction of B19V with Gb4.** RBCs (0.5% in 100 μl PiBS) were incubated with B19V ($5x10^9$) at pH 7.4 or 6.3 at different temperatures for 1h. The cells were subsequently washed at room temperature or at 4°C and viral DNA was extracted and quantified by qPCR.
(TIF)

**S8 Fig. Preincubation of viruses or RBCs separately at acid pH does not support hemagglutination at neutral pH.** B19V ($5x10^9$) and RBCs (0.5% in 100 μl PiBS) were incubated separately at acidic pH for 1h. Subsequently, the HA was performed at neutral (7.4) of acidic (6.3) pH. HA, hemagglutination assay.
(TIF)

## Acknowledgments

We are grateful to Beatrice Frey for the assistance with the transmission electron microscope.

## Author Contributions

**Conceptualization:** Jan Bieri, Remo Leisi, Carlos Ros.

**Data curation:** Jan Bieri, Carlos Ros.

**Formal analysis:** Jan Bieri, Remo Leisi, Carlos Ros.

**Funding acquisition:** Carlos Ros.

**Investigation:** Jan Bieri, Carlos Ros.

**Methodology:** Jan Bieri, Remo Leisi, Cornelia Bircher, Carlos Ros.

**Project administration:** Carlos Ros.

**Resources:** Carlos Ros.

**Supervision:** Carlos Ros.

**Writing – original draft:** Jan Bieri, Carlos Ros.

**Writing – review & editing:** Jan Bieri, Remo Leisi, Cornelia Bircher, Carlos Ros.

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
