## [Decision Letter · Decision Letter 0]

30 Mar 2021

Dear Dr. Ros,

Thank you very much for submitting your manuscript "Human parvovirus B19 interacts with globoside under acidic conditions as an essential step in endocytic trafficking" for consideration at PLOS Pathogens. As with all papers reviewed by the journal, your manuscript was reviewed by members of the editorial board and by several independent reviewers. The reviewers appreciated the attention to an important topic. Based on the reviews, we are likely to accept this manuscript for publication, providing that you modify the manuscript according to the review recommendations.

Sincerely,

Patrick Hearing

Associate Editor

PLOS Pathogens

Erik Flemington

Section Editor

PLOS Pathogens

Kasturi Haldar

Editor-in-Chief

PLOS Pathogens

orcid.org/0000-0001-5065-158X

Michael Malim

Editor-in-Chief

PLOS Pathogens

orcid.org/0000-0002-7699-2064

Reviewer Comments (if any, and for reference):

Reviewer's Responses to Questions

**Part I - Summary**

Reviewer #1: The manuscript “Parvo B19 interacts with globoside under acidic conditions to facilitate endocytic trafficking” by Bieri, Leisi, Bircher and Ros is aa comprehensive and elegant study that redefines the role of Gb4 in infection of cells by B19. Gb4 was previously thought to be the principle cell surface receptor for B19 based on virus binding to Gb4 and the fact that Gb4 knockouts are resistant to infection. The broad cellular distribution of Gb4 however is counter to the restricted tropism of the virus. The authors show that rather than playing a role in initial binding to cells that Gb4 plays a fundamental role downstream of initial entry to facilitate endocytic trafficking of the viral particle and its genome to the nucleus to initiate infection. Another cell surface molecule identified only as VPu-Receptor is the major cell surface receptor for the virus and its expression is highly restricted to erythroid progenitor cells (EPC). This paper focuses on the role of Gb4, confirms its interaction with B19 capsids, and defines the cellular requirements that dictate those interactions.

In summary this is an elegant and comprehensive paper, the data are clean and convincing, and certainly support their major conclusions.

Reviewer #2: Bieri and colleagues provide compelling evidence that a specific sphingoglycoprotein, designated Gb4, is a host factor for parvovirus B19 entry into erythroid progenitor cells. Although the principal attachment receptor for this virus has yet to be identified, it appears that cell tropism is largely determined by recognition of Gb4 that presumably interacts with the VP2 capsid protein. The authors show that this interaction does not occur at neutral pH, thereby explaining how the broad distribution of Gb4 does not result in infection of non-productive host cells. Rather the Gb4-B19 virus interaction occurs at a reduced pH, similar to that found in endocytic vesicles. Abrogation of Gb4 in cells deficient of this molecule results in efficient virus attachment and internalization, however the virus fails to progress and remains trapped in vesicles. The results are quite clear as presented and add a new dimension to our understanding of how multiple host cell molecules facilitate virus entry. I have only a few suggestions for the authors to consider in order to clarify a few sections of the paper.

1. The authors could bolster the Discussion a bit with regards to the identity of the unknown attachment receptor. For example, do they know the relative binding affinity for the putative receptor vs that for Gb4?

2. Figure 5C. I cannot find a description of the results of this experiment in the text. I think it may have been omitted accidentally by the authors.

3. Why doesn’t the interaction of the virus with Gb4 at low pH result in capsid protein production given that NS1 mRNA is detected at a normal level as shown in Figure 6D.

4. Line 248. I suggest a modification of the title as follows: The early endosomal compartment provides optimal conditions for Gb4 interaction.

5. The Material and Methods are lacking in sufficient detail in a few places:

Line 469. Please specify the source of the B19 used in this study (is this a clinical specimen or one purchased, etc)?

Line 513. Please give more details of the confocal microscopy.

Line 535. How was viral NS1 mRNA quantified and what primers may have been used?

Reviewer #3: The manuscript by Bieri et al., describes, in a comprehensive set of experiments, the role of globoside in the life cycle of human parvovirus B19. This laboratory has made seminal contributions in the past relating to B19 biology, and the current studies are a fine example of their thoughtful rationale, careful planning of experiments, and lucid presentation of the data. Overall, this is a well-written manuscript in which the quality of the data presented is good, and the conclusions drawn are generally supported by the data shown.

**Part II – Major Issues: Key Experiments Required for Acceptance**

Reviewer #1: None

Reviewer #2: (No Response)

Reviewer #3: None.

**Part III – Minor Issues: Editorial and Data Presentation Modifications**

Reviewer #1: The abstract on the manuscript submission form differs slightly from the abstract in the manuscript and the former has some confusing typos such as:

1st sentence: The glycosphingolipid (GSL) globoside (Gb4) this study, we applied artificial viral

particles, genetically modified cells, and specific competitors to address the interplay

between the virus and the GSL.

Figure 1A and S2 don’t seem to fit with the results:

“The expression of VP1uR in cells was sufficient to trigger virus attachment and uptake, however, in the absence of Gb4, the intracellular capsids failed to initiate the infection (Figs 1A and S2)”.

These figures are not showing failure to infect…1A is detection of receptor in GB4 KOs and S2 is showing cell cycle arrest. Please clarify.

Reviewer #2: Include a description of the studies shown in FIgure 5C.

Reviewer #3: The authors may wish to address the following points, which may be helpful to the reader:

1. While it is known that exposure to low pH alone is sufficient for the structural change observed in the adenovirus capsid, previous studies with AAV have suggested that exposure to low pH is necessary but not sufficient (J. Virol., 75: 4080-4090, 2001). Since B19 would also be expected to encounter similar intracellular milieu, in addition to pH changes, which may not directly mimic the in vitro experimental conditions.

2. Parvovirus B19 binding to RBCs in vitro may also not truly mimic a natural infection in vivo. Weigel-Kelley et al, have proposed that human parvovirus B19 binds to mature erythrocytes, which facilitate trafficking of the virus to the bone marrow where the target cells reside (Blood, 102: 3927-3933, 2003). This reference should be cited.

PLOS authors have the option to publish the peer review history of their article (what does this mean?). If published, this will include your full peer review and any attached files.

Reviewer #1: No

Reviewer #2: No

Reviewer #3: No

Figure Files:

Data Requirements:

Reproducibility:

References:

---

## [Editor Report · Decision Letter 1]

12 Apr 2021

Dear Dr. Ros,

We are pleased to inform you that your manuscript 'Human parvovirus B19 interacts with globoside under acidic conditions as an essential step in endocytic trafficking' has been provisionally accepted for publication in PLOS Pathogens.

Best regards,

Patrick Hearing

Associate Editor

PLOS Pathogens

Erik Flemington

Section Editor

PLOS Pathogens

Kasturi Haldar

Editor-in-Chief

PLOS Pathogens

orcid.org/0000-0001-5065-158X

Michael Malim

Editor-in-Chief

PLOS Pathogens

orcid.org/0000-0002-7699-2064
---

## [Editor Report · Acceptance letter]

16 Apr 2021

Dear Dr. Ros,

We are delighted to inform you that your manuscript, "Human parvovirus B19 interacts with globoside under acidic conditions as an essential step in endocytic trafficking," has been formally accepted for publication in PLOS Pathogens.

Best regards,

Kasturi Haldar

Editor-in-Chief

PLOS Pathogens

orcid.org/0000-0001-5065-158X

Michael Malim

Editor-in-Chief

PLOS Pathogens

orcid.org/0000-0002-7699-2064